# Wi-PoS: A Low-Cost, Open Source Ultra-Wideband (UWB) Hardware Platform with Long Range Sub-GHz Backbone

**DOI:** 10.3390/s19071548

**Published:** 2019-03-30

**Authors:** Ben Van Herbruggen, Bart Jooris, Jen Rossey, Matteo Ridolfi, Nicola Macoir, Quinten Van den Brande, Sam Lemey, Eli De Poorter

**Affiliations:** IMEC, IDLab, Department of Information Technology, Ghent University, 9000 Ghent, Belgium; Bart.jooris@ugent.be (B.J.); Jen.rossey@ugent.be (J.R.); Matteo.ridolfi@ugent.be (M.R.); Nicola.macoir@ugent.be (N.M.); Quinten.vandenbrande@ugent.be (Q.V.d.B.); Sam.lemey@ugent.be (S.L.); Eli.depoorter@ugent.be (E.D.P.)

**Keywords:** UWB, indoor localization, open source, hardware, shield, DW1000, Zolertia RE-Mote, ranging, external antenna, sub-GHz

## Abstract

Ultra-wideband (UWB) localization is one of the most promising approaches for indoor localization due to its accurate positioning capabilities, immunity against multipath fading, and excellent resilience against narrowband interference. However, UWB researchers are currently limited by the small amount of feasible open source hardware that is publicly available. We developed a new open source hardware platform, Wi-PoS, for precise UWB localization based on Decawave’s DW1000 UWB transceiver with several unique features: support of both long-range sub-GHz and 2.4 GHz back-end communication between nodes, flexible interfacing with external UWB antennas, and an easy implementation of the MAC layer with the Time-Annotated Instruction Set Computer (TAISC) framework. Both hardware and software are open source and all parameters of the UWB ranging can be adjusted, calibrated, and analyzed. This paper explains the main specifications of the hardware platform, illustrates design decisions, and evaluates the performance of the board in terms of range, accuracy, and energy consumption. The accuracy of the ranging system was below 10 cm in an indoor lab environment at distances up to 5 m, and accuracy smaller than 5 cm was obtained at 50 and 75 m in an outdoor environment. A theoretical model was derived for predicting the path loss and the influence of the most important ground reflection. At the same time, the average energy consumption of the hardware was very low with only 81 mA for a tag node and 63 mA for the active anchor nodes, permitting the system to run for several days on a mobile battery pack and allowing easy and fast deployment on sites without an accessible power supply or backbone network. The UWB hardware platform demonstrated flexibility, easy installation, and low power consumption.

## 1. Introduction

Ultra-wideband (UWB) has been a hot topic in the indoor localization, as UWB transceivers compatible with the IEEE 802.15.4a standard are becoming more accessible. The short UWB pulses result in a high time-domain resolution in the order of nano- or picoseconds, enabling centimeter-precision localization with excellent immunity against multipath fading [1]. In addition, the high bandwidth (>500 MHz) of the technology results in a high channel capacity, according to the Shannon–Hartley theorem [2], leading to promising prospects in terms of data rate and power consumption. Furthermore, the high channel capacity permits low transmission powers, avoiding narrowband interference with existing wireless technologies.

Although this is very promising, ongoing research remains bothered by several limitations on a variety of different aspects of the localization. Many different use cases for UWB localization systems are present, part of ongoing research, and are targeted by commercial applications. However, one of the main shortcomings in building a suitable UWB localization system is the availability of high-performance and cost-effective open source hard- and software, enabling the construction of flexible and modular testing environments. To compensate for this shortcoming, we developed Wi-PoS, a new hardware platform for UWB localization that offers several unique features and is publicly (open source) available. Wi-PoScombines different wireless technologies to achieve a full wireless system for accurate positioning. The hardware platform is designed for minimal pulse distortion and optimal ranging results. Distortion of UWB pulses, which should be minimized for optimal performance, can be attributed to four main causes: multipath propagation, noise, frequency dispersion, and the radio frequency (RF) front end of the printed circuit board (PCB). In Figure 1, the UWB link is given, together with the main causes of pulse distortion and a table that shows at which place in the link the distortions appeared. The RF front end is of utmost importance in the development of an UWB localization hardware platform. Hence, during the design of the hardware platform, extreme precautions were taken to limit the pulse distortion caused in the RF front end. The design of the RF front end for these high bandwidth pulses is challenging, and only a limited amount of commercial systems offer UWB localization platforms that interface with external antennas. Each of these commercially available systems exhibits severe limitations: not open source, no long-range back-end communication, expensive, and no network stack.

The main unique features of the Wi-PoShardware platform are as follows:
Flexible interfacing with external UWB antennas gives the opportunity to optimize the antenna system for the intended use case. The localization of assets can benefit from the use of dedicated impulse radio (IR)-UWB antennas [3,4,5]. The performance of different antenna designs can be evaluated with the flexible antenna interface.Three complementary wireless technologies are supported: UWB, 2.4 GHz, and sub-GHz. While UWB is used for very accurate localization, the lower power consumption, long-range sub-GHz technology can be used for communication and MAC level synchronization. The sub-GHz communication between nodes allows the development of a localization system with a completely wireless backbone, similar to References [6,7], where UWB is combined with a WiFi backbone. Although 2.4 GHz communication is not used in the current implementation, it is available in the platform when designing specific use cases.The hardware platform guarantees excellent pulse properties by minimizing pulse distortion. Thereto, the RF design was carried out in a full-wave simulator to ensure excellent impedance matching. Moreover, the selection of PCB laminates and components was carried out with utmost care and electromagnetic interference was minimized by means of via stitching.The hardware platform is compatible with the Contiki OS and openWSN, allowing reuse of existing IPv6 IoT protocol stacks for UWB system designs, as well. The Time-Annotated Instruction Set Computer (TAISC) framework [8] provides an efficient way to implement the network stack and gives the opportunity to research different MAC designs and easily adjust the MAC layer to different use cases.The hardware devices can be configured as both anchor and tag node to build a full localization system.


The main contributions of this paper are as follows:
The hardware platform is provided as open-source code allowing easy integration in other projects. The source-code of the UWB hardware and MAC protocol software is provided as open-source contributions [9].The hardware platform is thoroughly evaluated demonstrating superior range and accuracy with an extremely low power consumption.Guidelines in addition to the DW1000 application notes are given, describing design optimizations which allow other designers to optimize their UWB hardware solutions.


The remainder of this paper is organized as follows: Section 2 discusses relevant related work and alternative hardware boards. Section 3 provides an overview of the most important design choices and the technical specifications of the hardware platform, along with an introduction on the integration of Contiki-NG OS and the TAISC [8] software platforms. Section 4 presents and discusses experimental results that were achieved with the Wi-PoShardware platform. Section 5 summarizes the work and presents the concluding remarks.

## 2. Related Work

### Open Source Hardware

A plethora of UWB localization systems already exist. However, the majority is commercialized and not intended for scientific research purposes. This paper mainly focuses on open source systems. An overview of these systems is given in Table 1. The EVB1000 board from Decawave is also included in the comparison, as this board is specifically developed for research purposes and is purchasable in kits of four devices for $990.

The PolyPoint project [10] introduced multiple antennas to avoid the effect of polarization mismatch between sender and receiver. The Atlas project [11] designed a UWB node starting from the DWM1000 [12], in combination with the STM32F103C8T6 Evaluation Board known as “Blue Pill”. Both the hardware [13] and accompanying localization software [14] are open source. Different drivers for the DW1000 and DWM1000 [12] are written and are publicly available on GitHub [15,16]. The GitHub library [17] contains hardware design files.

The unique features of Wi-PoS, emphasized in the introduction, are included in Table 1. The majority of the hardware solutions is based on the DWM1000 module with integrated surface mount dielectric (SMD) chip antenna and does not support flexible interfacing with external antennas. The use of an external antenna inherently complicates the design of the RF front end, but many use cases can benefit from it. Mobile nodes typically require omnidirectional antennas, while anchor nodes can benefit from directional antennas. The influence of the integration platform will be minimized during antenna design, requiring tailoring for specific use cases. The only localization system that also includes this feature is the evaluation board from Decawave (EVB1000). The integration of sub-GHz is a unique feature. Some other UWB localization solutions use an extra radio, but with a shorter range wireless technology. The range of sub-GHz (>1.5 km) is significantly bigger than the WiFi range (>150 m), improving the scalability of the localization system [19] and developing localization systems with large coverage areas. A mobile tag can communicate with a central routing node at large distances, allowing the central node to control the localization schemes [7].

The presented hardware platform was evaluated on different performance characteristics. Several research papers have already been published about the comparison of different UWB localization systems. The Bespoon, Decawave, and Ubisense systems are compared in Reference [20], while other published work comparing different UWB localization systems, includes Reference [21,22]. These comparative studies mainly focus on the achieved ranging accuracy in different operating conditions, but they use different ranging techniques (angle of arrival, time of flight), or the various UWB localization systems are compared in different measurement environments. The general conclusion the researchers take is that the Decawave system is slightly better in line-of-sight (LOS) conditions and more reliable in non-line-of-sight (NLOS) than the other systems (Bespoon and Ubisense). In this paper, we will evaluate the accuracy that was achieved with the developed hardware, together with the energy consumption and range of the platform, as these parameters are important for deployments with mobile battery packs, which was one of the targeted use cases of the platform. Specific attention was devoted to ensure the data were reproducible by minimizing the influence of environmental factors.

## 3. Design Approach

### 3.1. Conceptual Design

The hardware platform is made up of a UWB RF front end and a sub-GHz backbone (Figure 2). The RF front end consists of the DW1000, the necessary pheriperal circuits, and a power connection. For the sub-GHz backbone, an existing low-cost IoT board, the Zolertia RE-Mote [23], is used. The choice for the Zolertia RE-Mote for the backbone is based on several arguments. First, the board is an existing, of-the-shelf IoT platform with widespread software support from both Contiki OS and openWSN. The RE-Mote has on-chip support for two popular, low-power IoT transceivers: the CC1200 long-range, sub-GHz communication (820–950 MHz) and the CC2538 for low-power, short range communication in the 2.4 GHz bands. Furthermore, the TAISC framework is available for the RE-Mote, and it supports SPI communication with the DW11000.

Although the layout of the UWB shield (Figure 3) has been designed for the RE-Mote, it also interfaces with other hardware platforms (Arduino, other IoT boards) by connecting the correct pins. Moreover, the hardware design is provided as open-source code, allowing easy modification towards other platforms. The full specifications of the UWB radio board can be found in Table 2. All pins of the Zolertia RE-Mote are connected to the Wi-PoSshield permitting future expansion of the system with additional sensor components.

#### Optimization Techniques to Minimize Pulse Distortion

As discussed earlier in the introduction of the paper, designing a board for such large bandwidths with minimal pulse distortion is challenging. An optimal board design is optimized for all six frequency bands supported by the DW1000 UWB transceiver.

For this reason, the Wi-PoShardware includes the following optimizations:
A 4-layer stack up is used for the PCB (Figure 4). The 2nd and 3rd layer are used as the ground plane and power plane, respectively. The use of a ground plane reduces voltage drops in potential of ground level throughout the board and creates short return paths for the currents from the different components, especially for the high sensitive signals on the top layer like the transmission lines. The ground layer shields the sensitive components from different noise sources on lower layers. A separate analog and digital ground is used, joined in the middle of the board.The PCB has a thickness of 1.568 mm and is fabricated with high-quality Rogers 4350 (RO4350) material for optimal RF behavior between the top layer and layer 2 and between bottom layer and layer 3 (Figure 4). The dielectric constant for this material is well specified and dielectric losses are significantly smaller than with the frequently used FR4 material, especially for higher frequency use cases. Furthermore, the permittivity for RO4350 is more stable and the FR4 will have more fluctuations in permittivity between different batches.The path from DW1000 towards the SMA connector is a straight line shielded by stitching vias to prevent electromagnetic interference and minimize pulse distortion. The widths of the microstrip lines are matched with the desired impedance: 100 Ω for the differential pair from DW1000 to balun and 50 Ω for the single transmission line from the balun to the single ended SMA connector. Care is taken that no obstacles are placed on the return path at the ground layer.No components are placed sideways of the path to the antenna, no signal whatsoever is routed in that area of the board, no perpendicular angles are taken in the paths, especially for sensitive high frequency signals, and the decoupling capacitors with the smallest values are placed as close as possible to the DW1000 on the power track.


These design optimizations can also be used for the design of other, future UWB boards. The effect of this optimization is profound, as will be shown in later sections.

### 3.2. Open Source Network Stack

Many commercial products exist today, but these do not provide access to their firmware, thereby limiting their potential for research purposes. In contrast, a number of open source academic solutions is available (see Section 2). However, as discussed, existing platforms mainly provide source code for the localization algorithm, omitting the code for MAC protocol design, and higher layer networking protocol stack for large scale wireless deployments. Our platform provides a full IPv6 compatible IoT protocol stack, including a MAC design framework for designing multi-technology MAC protocols and multi-hop mesh networking protocols.

#### 3.2.1. Localization Algorithm

For the position calculation, a particle filter implementation is available by default, based on the work of Reference [24]. The particle filter uses a set of N particles that represents different possible states. The states of each particle are updated with each incoming range, resulting in a position update for every reporting message. The particle server requires incoming ranges from at least three different anchors for a good 2D position estimate and at least four different anchors on different heights for a 3D estimate.

Every incoming position estimate is published as an MQTT [25] resource, which is available either on the mobile tag or transmitted through one of the radio interfaces to a back-end server. By subscribing to the relevant MQTT topic, new localization algorithms can easily be implemented. Similarly, each position estimate is published as an MQTT topic, allowing easy integration with existing systems.

#### 3.2.2. Multi-Technology MAC Design Framework

For the design of the MAC protocol, our board supports the TAISC framework [8], which is designed to simplify the design of time-critical MAC protocols. The TAISC framework provides interfaces towards all three supported radio chips (IEEE 802.15.4 2.4 GHz, IEEE 802.15.4 g sub-GHz, and IEEE 802.15.4 UWB). Each interface can run an independent MAC protocol, or novel multi-interface MAC protocols can be designed. Example MAC protocols that are provided include CSMA/CA, ALOHA, slotted ALOHA, IEEE 802.15.4e TSCH, ContikiMAC TODO:ref, and several TDMA variants. In addition, a multi-interface TDMA MAC implementation is available that combines the strengths of two radio interfaces: sub-GHz for reporting and TDMA slot allocation, and UWB for ranging (see Section 4).

#### 3.2.3. Contiki OS

Finally, the board supports Contiki-NG [26], an open-source operating system for next generation IoT devices. The operating system is designed for networked, low power IoT devices. To this end, it supports IPv6, along with standardized IETF low-power wireless standards: 6LoWPAN, RPL, CoAP, and OMA LWM2M. All of these protocols can be run on top of all support radio platforms, allowing the design of heterogeneous mesh protocols that utilize UWB, sub-GHz, and 2.4 GHz communications in parallel.

## 4. Evaluation

### 4.1. Test Setup

The Wi-PoShardware platform combines several unique features with centimeter-precision localization. The performance of the localization system is evaluated on ranging accuracy and power consumption. First, the receiver sensitivity, link margin and packet receive ratio are measured for the Wi-PoShardware platform. Next, the accuracy of the system is presented. The additional sub-GHz backbone communication offers supreme reductions in power consumption at the anchor nodes, compared to solutions that use UWB for communication between the nodes. In all experiments, we measured at channel 1 with a 110 kbps data rate, 64 MHz PRF , and preamble of 1536 symbols. The transmit gain is 12.5 dB, as this is the default value of the Decawave DW1000 transceiver for the chosen channel and PRF. If other parameters are used, it will explicitly be stated.

To calculate the distance between the 2 nodes, asymmetric two way ranging (TWR) [27] is used, as this cancels out clock drift in sender and receiver as well as different reply times between messages [12]. With this protocol (Figure 5), we can use the same poll message for multiple anchor nodes. During 1 superframe, the tag will sequentially range with four different anchors, where every anchor gets a specific timeslot in the superframe appointed for ranging with the tag node. At the start of the superframe, the tag sends a sub-GHz beacon with the anchors he desires to range with and upon reception, the anchor nodes will allocate time slots for transmit and receive messages with the tag. Then, the tag will send an UWB poll message and every anchor will answer in his dedicated timeslot with a response message. The tag will send a final message to the anchor node. After receiving this final message, the distance between anchor and tag is calculated at the anchor node and reported over the sub-GHz backbone to a server and to the tag node. Every ranging requires three consecutive UWB messages to be transmitted and received successfully. When a packet is not correctly received, no retransmissions will be sent. An in depth study of the protocol can be found in Reference [28].

The DW1000 timestamps the packet when it enters the differential input. The time for the signal to propagate between reception at the antenna, and entering the DW1000 is compensated with the *Antenna Delay* value. This value contributes to a constant ranging error, and it is required to be calibrated for each device for optimal results.

### 4.2. Receiver Sensitivity, Link Margin, and Packet Receive Ratio

The ability of the receiver to detect arriving packets is mainly influenced by the power level of the incoming signal. The receiver sensitivity, the lowest power where the receiver is able to successfully receive a packet, is influenced by many ranging parameters, (ranging channel, preamble, PRF, etc.). To determine the receiver sensitivity for the used set of ranging parameters, a UWB link is set up between an anchor module and a tag module, with a programmable attenuator, the quadAtten from octoBox [29], in between them. The test setup is given in Figure 6. The dynamic range from the attenuator is 0 to 55 dB. An extra attenuation of 30 dB is added to the test setup so path attenuations from 30 to 85 dB could be measured.

For the Wi-PoShardware, the reported receive power, packet receive ratio, and range success ratio are measured for different attenuation levels and given in Figure 7. Ranging fails when one out of the three packets in the asynchronous two-way ranging scheme is lost, and the ranging starts to fail whenever the received power is less than −100 dB m (path attenuation of 85 dB), but at −106 dB m still 50% of the ranges succeed. As expected, the ranging capabilities degrades faster than the packet reception as ranging requires all three of the packets to be received. The lowest receive power reported where packets are detected (>1% of the packets are received) is at −109 dB for both boards.

### 4.3. Ranging Accuracy

#### 4.3.1. Theoretical Simulation Model

A simplified theoretical model for the ranging is derived based on the literature for UWB propagation channels [12,30,31]. This model can be used to understand the different effects that occur during ranging, and to explain and predict the ranging results. The theoretical model is a combination of the path loss model and the ground reflection model. The path loss model accounts for the losses during propagation in the air for the signal, and the ground model includes the interference between the direct path and the most important reflection with the ground. Based on this analysis a simulation curve is determined to predict and interpret the experimental results.

**Path loss model:** The magnitude of the degeneration of the signal strength during propagation can be dedicated to different influences: the transmit power of the sender ($P_{t}$), the gain of the antennas at sender ($G_{t}$) and receiver ($G_{r}$), the losses during propagation at the PCB ($L_{t}$ and $L_{r}$), and the losses during propagation of the signal through the air (PL) depending on the length of the transmission path between sender and receiver. The link budget of the signal during propagation for the LOS path can be evaluated by using Friis path loss formula:
(1)$$P_{r}=P_{t}+G_{r}+G_{t}-L_{r}-L_{t}-PL.$$


The losses in the PCB and the gain of the antennas will be the same for different distances/environments and are hardware specific. During the evaluation of the experiments an approximation of these factors can be made. The link margin is expressed as the difference between the receiver’s sensitivity and the expected minimum receive power. The link margin should be high enough for correct ranging. Equation (Equation 1) points out a dependence between receive power and transmit power. The link margin can be increased by increasing the transmit power. The transceiver has a maximal value of 33.5 dB in steps of 0.5 dB for the transmission power.

The path loss component from Equation (Equation 1) is dependent on the distance between sender and receiver, the frequency, and the propagation speed. This path loss component is based on two components, a fixed value measured at the reference distance $d_{0}$ = 1 m and a logarithmic factor on the relative increase in distance between sender and receiver.
(2)$$PL\left(d\right)=PL_{0}\left(d_{O}\right)+10\;\gamma \;log_{10}\left(\frac{d}{d_{0}}\right)+X_{\sigma };d>d_{0},$$
where the $PL_{0}$ factor is the path loss at a distance of 1 m, and γ is the path loss exponent which has value 2 for the free space situation and is chosen 1.58 in the outdoor measurements as suggested value for farm (outdoor) environments in Reference [32]. The $X_{\sigma }$ factor is zero-mean Gaussian random variable with standard deviation σ and this factor can be referred to as shadowing and captures the path loss deviation from its median value.

**Ground reflection model:** Interference will occur at the receiving antenna between the direct, LOS, path, and other (reflected) paths with different lengths from sender to receiver. In the ground reflection model, the interference between the most important reflection path and the direct path is taken into account (see Figure 8). The difference in path length will result in distances where both signals will interfere constructively, increasing the receive power, or destructively, where they cancel each other out. The separations between sender and receiver where the signals interfere destructively are dependent on the center frequency of the UWB ranging and the height of the antenna nodes.

#### 4.3.2. Indoor Short Range Test

The accuracy of the Wi-PoShardware system is tested in indoor lab environment conditions for small distances (between 1 and 5 m). The nodes are placed on tripods at a height of 1.5 m. The groundtruth distances for this measurement are determined with a 1.5 mm-accuracy laser meter, and the results of the experiment can be found in Figure 9b. For these small distance indoor experiments, the influence of multipath effects is rather small as the reflected waves path length is significantly larger than the line of sight path. The accuracy of these measurements are below the 10 cm level.

#### 4.3.3. Outdoor Long Range Test

Longer range measurements were performed on an empty parking lot. Test samples were recorded at two fixed distances (50 and 75 m) at 2 m height, and the cumulative distribution of these experiments is given in Figure 9a. The statistics of these measurements are given in Figure 9b. The mean reported distances are 49.99 and 74.95 m, deviating only 1 and 5 cm from the real distance. The mean receive power at 75 m is −101.08 dB m, resulting in a packet receive ratio of 91.55%. The precision for the measurements at 1, 2, 5, and 50 m is very similar while the precision for 75 m is much lower due to the small link margin at 75 m. Ranging in indoor environments are typically more challenging than in open space. Reflections from ground, walls, objects, and ceiling will interfere at the receiver’s antenna influencing the signal strength level of the received packet.

A second outdoor measurement was carried out, where the distance between sender and receiver was gradually increased when registering receive power, first path power, and reported distance until a distance of 100 m was reached. The anchor and tag nodes were placed at a height of 2 m. The receive powers and reported ranges are displayed in Figure 10, and simulation results based on the theoretical model (Section 4.3.1) were added. Ranges were received until the previously measured power level (Section 4.2) is reached −106 dB, and packets with lower receive powers (−106 to −109 dB m) still were received, but no ranging could be determined. The simulation predicted that ranging was possible between 1 to 81.55 m and from 116.25 up to 220.35 m. If the transmit power gain of the DW1000 is increased to the maximal value, the simulation predicts UWB range until 1008.95 m and communication until 1227.65 m when the anchors are placed 2 m above ground. Adjusting the height will change the length of the ground reflected path and influences the range. The sub-GHz backbone permits to report the ranges from anchor to tag node over large distances.

### 4.4. Energy Measurements

One of the critical properties of a mobile localization system is the power consumption. For the Wi-PoShardware, the total power consumption is measured over the USB-C connector when the node is used as tag and as anchor. The current is measured with the N6705 DC power analyzer system [33] at a 10.24 μs sampling interval. A typical current profile for a tag node is given in Figure 11. When the device is in IDLE mode, the current flowing into the node is 58.26 mA, where changing the state of the DW1000 from IDLE to INIT can reduce to current to 44.26 mA. In the beginning of the superframe, the tag will send out a sub-GHz packet, followed by a UWB POLL message (broadcast to all four anchors). Next, one UWB packet is received, one UWB packet is sent, and one sub-GHz packet is received for each anchor before the superframe ends at 72.7 ms. Each anchor replies in its specified time slot. The DW1000 at this tag node is first put in receive (RX) mode and afterwards in transmit (TX) mode. After completion of the UWB message handling, the sub-GHz report is sent, and a smaller peak in current consumption can be noticed. Anchor nodes that currently are not ranging with the tag put the UWB radio in the INIT state, and the node only listens on the sub-GHz spectrum, reducing power consumption in localization systems.

The amount of current that is consumed during the different phases of the TDMA chain is given in Table 3. Anchor nodes only range one time each superframe with the tag, permitting the anchor to enter INIT state and limiting the current of the DW1000 transceiver to only 4 mA. The sub-GHz backbone is used for transmitting range update information to the different nodes and to a central server node, as the sub-GHz transmission consumes less power compared to UWB transmission. The sub-GHz radio is used for listing for beacons sent out from the tag, as listening on the sub-GHz radio leads to enormous power reduction compared to always-on UWB listening anchor nodes. The uses of sub-GHz for the communication and reporting message results in a power reduction of 14.3% and 20.5% for tag and anchor node, respectively, in comparison to systems that use UWB for these messages.

If the transmit gain of the DW1000 is raised, the current consumption also increases and, therefore, the trade-off between longer range and lower power consumption has to be made when setting up the transmit power gain, taking into account the regulations on maximal transmit power.

The excellent timing scheduling of the TAISC platform makes sure that the DW1000 is only put in receive mode when actually required. The actual fraction of the time a node is in receive/transmit mode is given in Table 3 for tag and anchor nodes. The mean power consumption for different types of nodes is calculated and given in Table 4. Most of the time, the devices are in the IDLE or INIT state. If the anchor is not part of the set anchors the tag is ranging with, no UWB packets are sent and the power consumption is very low. Increasing location update rate is possible but will increase the power consumption of the system. When the nodes are powered with a 6000 mAh battery pack, the tag will be powered for 74 h and the anchor nodes for 99 h, allowing easy deployment. When the superframe length is raised to 1 s, so every anchor ranges at 1 Hz, the anchor nodes can endure almost 132 h.

The power consumption for the evaluation board provided by Decawave is also measured as a comparison value. When configured as an anchor node for the default application software, the average current consumption is 173.29 mA. The anchor node continuously listens on the UWB radio, waiting for packets. The tag node consumes 52.84 mA on average. The ranging algorithm is between one tag and one anchor at a ranging update rate of 3.6 Hz when ranging on channel 2. When the update rate on this example is increased, the power consumption of the tag will increase significantly.

### 4.5. Production Cost

The total cost for the production of 1 Wi-PoSnode (in a batch of 50) is 178.22 euros, where most of cost is due to the price of the Zolertia RE-Mote (90.16 euros) and the PCB printing and assembling (78.32 euro). Minor costs include the purchase of cables for powering (USB-C) and flashing the RE-Mote and the antennas. The installation cost for these devices is rather limited as the backbone is completely wireless with the sub-GHz communication.

## 5. Conclusions

The limitation of current feasible UWB localization systems encouraged us to develop a new low-cost open source hardware platform. The completely new hardware platform, based on Decawaves DW1000 transceiver, for UWB localization has several unique features (flexible antenna interfacing, support for long-range sub-GHz, and 2.4 GHz communication, easy MAC implementation), allowing fast deployment in different use cases. The combination of the accurate UWB ranging information with the reliable sub-GHz backbone network increased the scalability of localization systems and reduces power consumption at the anchor nodes compared to systems that only use UWB for both ranging and communication. The flexible antenna interface allows design of the antenna for different specific use cases. For example, anchor nodes mounted at the walls can hugely benefit from using directional antennas. Mobile nodes typically have an omnidirectional antenna. The low power consumption of the hardware platform permits to range multiple days on a mobile battery pack, allowing (temporary) deployments without an accessible power supply or backbone network.

The board was thoroughly evaluated on different ranging characteristics and, combined with the added features the system, really outperformed the existing open source system; in challenging indoor environments, the accuracy was below 10 cm for small distances. For outdoor measurements at 50 and 75 m accuracies of 1 and 5 cm were achieved. A theoretical model was composed to simulate path loss effects and the effect of the most important ground reflection. The energy consumption was heavily influenced by the update frequency of the ranging, but with an update rate of 55 Hz, the nodes could last a few day on standard mobile battery packs. The efficient timing of the software stack makes the proposed hardware a very competitive UWB localization system for fast and easy deployment.

Multiple improvements are still in development. With the efficient MAC protocol design, the update frequency for ranging is increased, as well as extra reductions in power consumption by putting the chips on the Zolertia RE-Mote or the DW1000 into sleep mode and self-calibration of the anchor nodes.

## Figures and Tables

**Figure 1 sensors-19-01548-f001:**
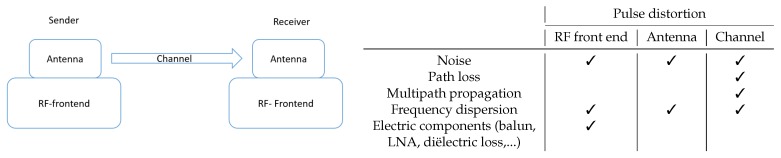
Ultra-wideband (UWB) link with the main causes of pulse distortion in a UWB localization system.

**Figure 2 sensors-19-01548-f002:**
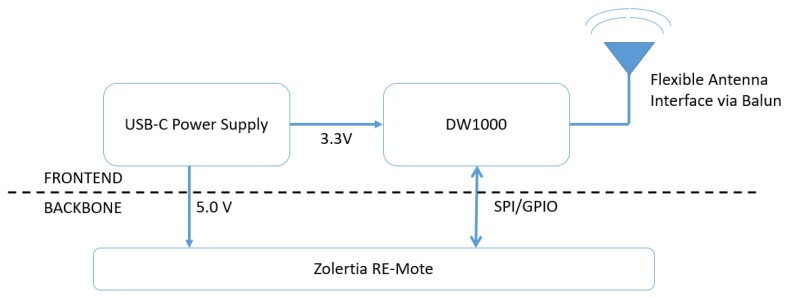
High-level architecture of the hardware platform.

**Figure 3 sensors-19-01548-f003:**
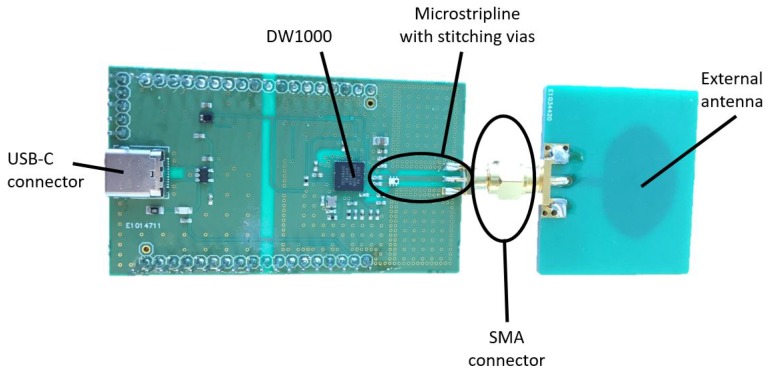
UWB shield and connected external omnidirectional antenna.

**Figure 4 sensors-19-01548-f004:**
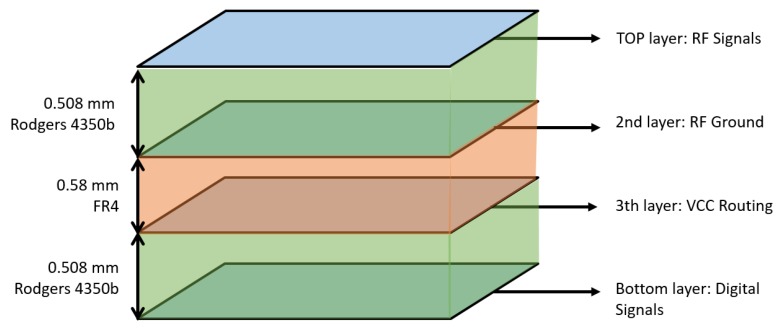
The four layers of the hardware PCB, their material, and their thickness.

**Figure 5 sensors-19-01548-f005:**
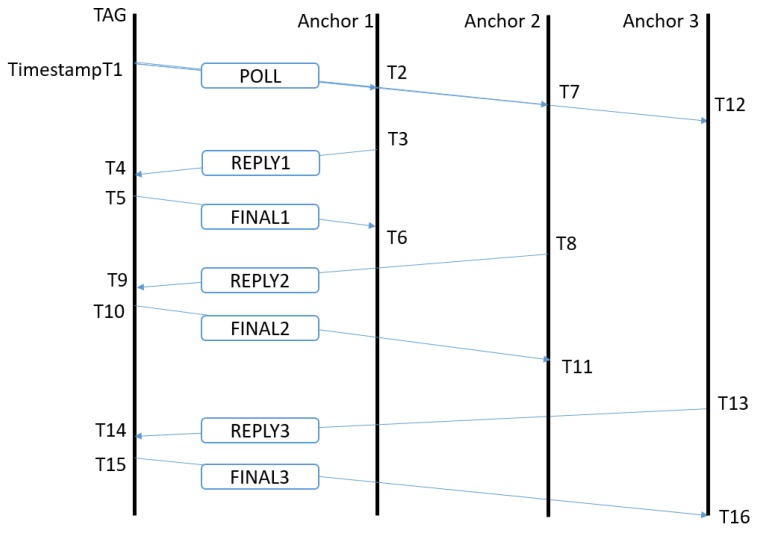
Asynchronous two-way ranging scheme of the UWB packets for a three-anchor system. Only one poll is sent from the tag and shared over the different anchor nodes.

**Figure 6 sensors-19-01548-f006:**
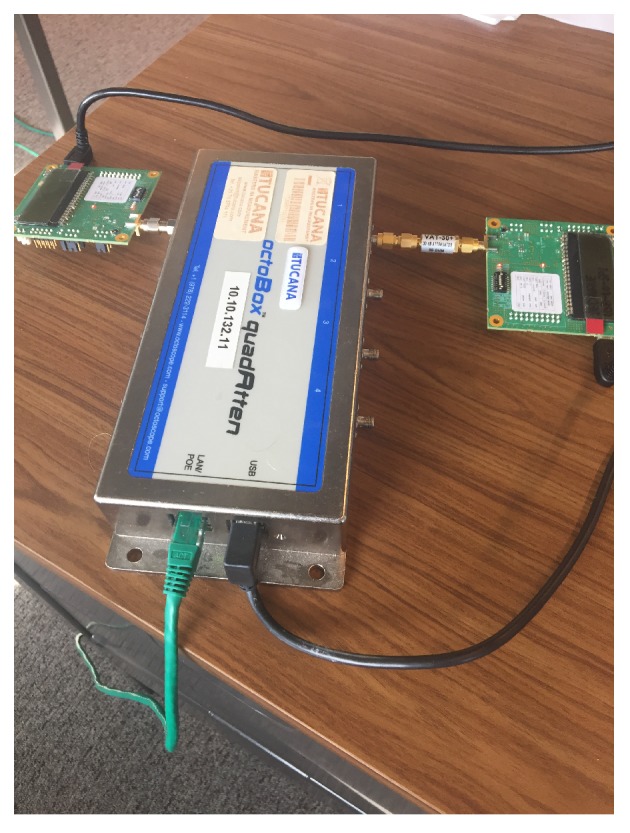
Test setup with attenuator on the path between sender and receiver.

**Figure 7 sensors-19-01548-f007:**
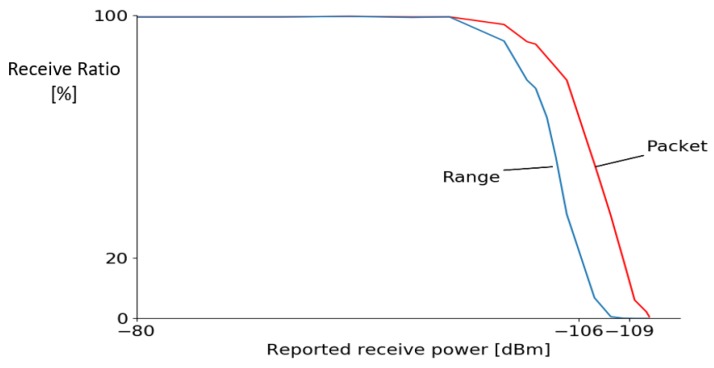
Packet/Range received ratio for different path attenuations and receive powers.

**Figure 8 sensors-19-01548-f008:**
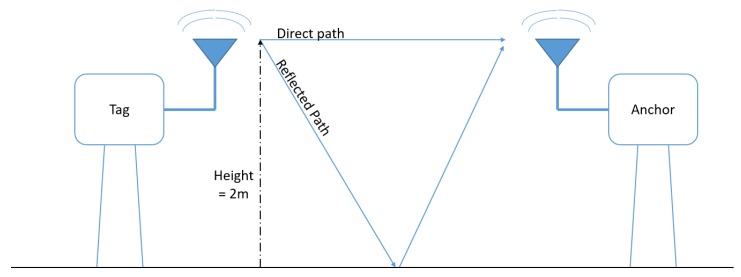
Ground reflection model.

**Figure 9 sensors-19-01548-f009:**
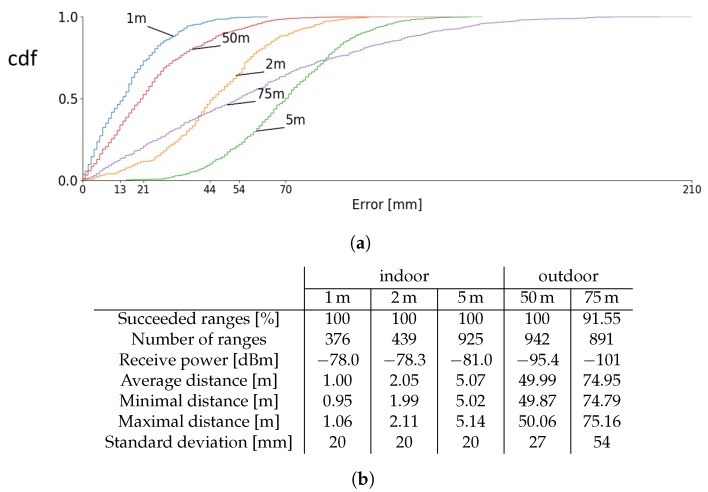
Indoor short distance and outdoor long distance measurements. (**a**) Cumulative distribution function of the reported ranges. (**b**) Statistics measurements.

**Figure 10 sensors-19-01548-f010:**
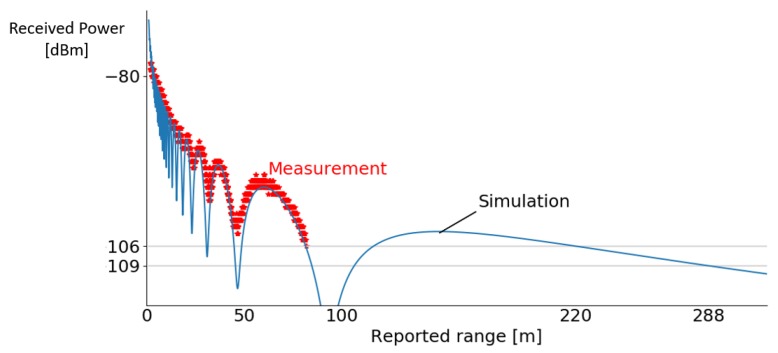
Reported received power at different distances + simulation curve from path loss and ground reflection model.

**Figure 11 sensors-19-01548-f011:**
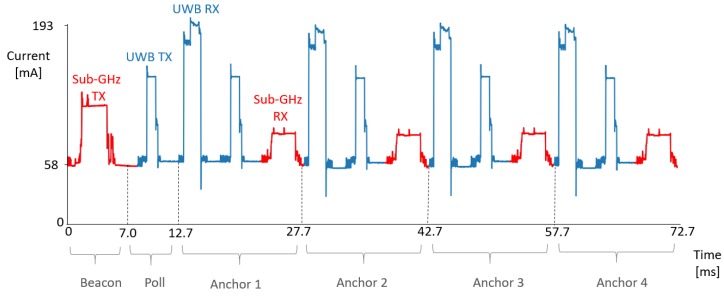
Current consumption in tag node.

**Table 1 sensors-19-01548-t001:** Overview of existing open source UWB localization systems. Wi-PoSsupports several extra radio modules, an external antenna interface, and Open Source Hardware, localization algorithm, and network stack.

Project	Academic/Commercial	External Antenna	Extra Radio	Open Source		
				HW	Localization Algorithm	IoT Network Stack
PolyPoint [10]	academic		**✓**(BLE)	**✓**	**✓**	
Atlas [13,14]	academic		**✓**	**✓**		
Arduino DW1000 [15]	academic				**✓**	
DecaDuino [16]	academic				**✓**	
KDWM1000 [18]	academic			**✓**	**✓**	
uwb_localization [17]	academic		**✓**(Arduino Mini)	**✓**		
EVB1000 [12]	commercial	**✓**				
Wi-PoS	academic	**✓**	**✓**(sub-GHz and 2.4 GHz)	**✓**	**✓**	**✓**

**Table 2 sensors-19-01548-t002:** Specifications of the UWB shield.

Dimensions	40.29 × 70 × 1.568 mm^3^
Stack	4 layers
Printed circuit board (PCB) material	RO4350B and PR2116
Antenna interfacing	SMA connector
UWB transceiver	DW1000
Crystal oscillator	402F38411CAR (CTS-Frequency Controls) [±10 ppm]
Balun	HHM1595A1
Power	USB-C (5.0 V)

**Table 3 sensors-19-01548-t003:** Fractions of time when the platform is in which state per superframe (72.7 ms).

	Current [mA]	Time Tag [ms]	Tag [%]	Time Anchor [ms]	Anchor [%]	Time Anchor (No-Slot) [ms]	Time Anchor (No-Slot) [%]
UWB RX preamble hunt	177	1.724	2.4	0.431	0.6	0	0.0
UWB RX	193	5.6	7.7	2.5	3.4	0	0.0
UWB TX	140	3.41	4.7	0.682	0.9	0	0.0
sub-GHz TX	106	4.2	5.8	2	2.8	0	0.0
sub-GHz RX	80	8	11.0	4.2	5.8	4.2	5.8
IDLE	58	49.766	68.5	62.9	86.5	68.5	94.2

**Table 4 sensors-19-01548-t004:** Average current for anchor/tag nodes for different configurations (55 ranges/s).

TX Gain	0 dB	12.5 dB	33.5 dB
tag	80.53 mA	81.19 mA	83.40 mA
anchor (1-slot)	63.16 mA	63.29 mA	63.73 mA
anchor (no-slot)	47.98 mA	47.98 mA	47.98 mA

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
