# Peer review of "Wi-PoS: A Low-Cost, Open Source Ultra-Wideband (UWB) Hardware Platform with Long Range Sub-GHz Backbone"

_sensors, 2019, doi:10.3390/s19071548_

Round 1
Reviewer 1 Report
This work is new, as far as I know, and the topic addressed is important,.
The writing needs a careful revision.
The x axis of figure 9a. is not clear.
It is not clear how the localization is done, and how it compares with other techniques from the literature.
Author Response
Dear reviewer, editor,
We wish to thank you for your useful review and your appreciation of our work. Please find
enclosed feedback about the changes we made to the paper according to your feedback.
Best regards,
Ben Van Herbruggen

Reviewer 2 Report
This paper introduces a novel hardware platform provided as open-source either at software and hardware level. My big concern about the platform is that the repository does not contain any source code, but two messages stating "The hardware schematics will be released soon" and "The software will be released soon". I suppose that the authors will publish all code once this paper is accepted, but the current status should have been mentioned in the peer-review version of the paper. Moreover, the authors should have indicated that the HW schematics will be also available in the repository. Finally, you mention Hardware source-code in the paper, did you refer to schematics or to the firmware of the developed hardware?
In my opinion the paper is clearly written and the objectives/novelties of this work are well identified. However, the related work section is not as complete as I expected. The introduced system is clearly catalogued as 'academic'. This label might crash with one of the contributions "The hardware platform is thoroughly evaluated demonstrating superior range and accuracy with a extremely low power consumption.". With that bold sentence one might think that you are targeting to compete with commercial applications and, therefore the label 'Academic' would not fit. Thus, I consider that a more comprehensive comparison, including more commercial UWB positioning systems, should have been included in the related work and initial comparison. Moreover, I also miss the deployment costs of those systems shown in Table 1.
The evaluation procedure and testing is at the level of a journal paper, but some clarifications are needed. For instance, Section 4.3.2 introduces the results for the indoor short range test. Some details about the evaluation system are provided but:
- How many localization attempts are you doing at each distance?
- "this measurement are determined with a mm-accuracy laser meter" Could you please specify the error of the range measurement device?
- Could you please give more discussion about the results? Why the errors are lower for a 50-m distance (outdoors) than for a distance of 1-m indoors?
- Please, introduce all tables and figures in the correct order.
- Can you provide the accuracy values of the other platforms and solutions?
It is remarkable the Energy Measurements, but I am not able to match the energy measurements values of your proposal and the Decawave Board "When configured as anchor node, the average current consumption is 173.29 mA. The anchor node continuously listens on the UWB radio, waiting for packets. The tag node consumes 52.84 ms on average."
- Do you mean that DW consumes 173 ma and your platform around 80 ma? This means reducing the consumption to a half.
- "The tag node consumes 52.84 ms on average" ms o mA? If it is mA, your energy consumption is similar (slightly higher for 1-slot, slightly lower for no-slot).
The production cost is below 200€, which is also remarkable. Could you please indicate the cost of the other platforms and solutions to put the reader in the appropriated context?
English is fine but some minor typos should be fixed.
Author Response

(The authors gave the same response as above.)

Round 2
Reviewer 1 Report
The authors addressed the relevant issues.